# Versatile Approach of Silicon Nanofabrication without Resists: Helium Ion-Bombardment Enhanced Etching

**DOI:** 10.3390/nano12193269

**Published:** 2022-09-20

**Authors:** Xiaolei Wen, Lansheng Zhang, Feng Tian, Yang Xu, Huan Hu

**Affiliations:** 1Center for Micro and Nanoscale Research and Fabrication, University of Science and Technology of China, Hefei 230026, China; 2ZJUI Institute, Zhejiang University, Haining 314400, China; 3State Key Laboratory of Fluidic Power & Mechanical Systems, Zhejiang University, Hangzhou 310027, China; 4School of Micro-Nano Electronics, Zhejiang University, Hangzhou 310027, China

**Keywords:** nanofabrication, helium ion microscope, chemical etching, nanofluidic channel, nano-resonators

## Abstract

Herein, we report a helium ion-bombardment enhanced etching method for silicon nanofabrication without the use of resists; furthermore, we demonstrate its unique advantages for straightforward fabrication on irregular surfaces and prototyping nano-electro-mechanical system devices, such as self-enclosed Si nanofluidic channels and mechanical nano-resonators. This method employs focused helium ions to selectively irradiate single-crystal Si to disrupt the crystal lattice and transform it into an amorphous phase that can be etched at a rate 200 times higher than that of the non-irradiated Si. Due to the unique raindrop shape of the interaction volumes between helium ions and Si, buried Si nanofluidic channels can be constructed using only one dosing step, followed by one step of conventional chemical etching. Moreover, suspended Si nanobeams can be fabricated without an additional undercut step for release owing to the unique raindrop shape. In addition, we demonstrate nanofabrication directly on 3D micro/nano surfaces, such as an atomic force microscopic probe, which is challenging for conventional nanofabrication due to the requirement of photoresist spin coating. Finally, this approach can also be extended to assist in the etching of other materials that are difficult to etch, such as silicon carbide (SiC).

## 1. Introduction

For more than half a century, the capability to fabricate transistors on planar wafers with continuously smaller resolutions has been a major driving force for the development of integrated circuits. In addition to high resolution, the capability to fabricate nanostructures on irregular non-flat surfaces or surfaces with prefabricated micro/nanostructures is necessary for emerging applications, such as nano-electro-mechanical systems (NEMSs); furthermore, the fabrication of plasmonic nanostructures on an atomic force microscopic tip that allows the measurement of optical forces [1], bio-inspired super-adhesive surfaces [2], optical applications including biomimetic compound eyes [3], near-field tip-enhanced-Raman measurement [4], and quantum physics studies [5] is crucial.

The major technologies of top-down nanofabrication are photolithography and electron beam lithography, both requiring the use of a layer of photosensitive polymers for lithography, which presents a significant challenge in nanofabrication on nonplanar or irregular surfaces. Conventional spin coating cannot coat a uniform photoresist layer on non-flat surfaces. While spray coating [6] and Langmuir–Blodgett [7] techniques are applicable to some types of irregular surfaces, they cannot be used for surfaces with sharp corners or edges typically found in nanodevices. The method of evaporating photoresists suffers from low sensitivity [8], and the transfer of photoresist layers with a polymer stamp [9] requires several non-standard processing steps. Focused ion beam milling can be used to fabricate nanostructures directly on non-regular surfaces; however, its resolution is limited to dozens of nanometers.

Since its invention [10], helium ion microscopy (HIM) has demonstrated promising applications in nanofabrication [11,12,13,14,15,16,17,18], nano-metrology [19], and defect engineering [20] owing to its small beam spot (~0.5 nm) and minimal proximity effects. HIM utilizes a single-atom gas field ion source that provides high brightness to ensure sufficient beam currents even after passing through small apertures (<5 µm) [21]. HIM can also be used to generate defects for applications in electronics, including memtransistors [22], electrical tuning, and doping of 2D materials [23,24]. Recently, HIM was also employed for the 3D nanofabrication of photoresists [11] based on the unique 3D interaction volume between helium ions and photoresists, as well as for the fabrication of 3D nanostructures of Si and SiC based on He implantation-induced substrate swelling [25].

However, for single-crystal Si-based substrates, the use of HIM to directly produce Si nanostructures by milling has been a significant challenge; this is because focused helium ions typically induce surface deformation and lattice damage due to its low sputter yield and diffusivity [26], leading to effects such as subsurface nanobubbles and structure swelling [27]. All these defects and damage caused by helium ions considerably restrict the application of HIM in Si nanofabrication.

Herein, we report on the helium ion-assisted etching approach, which combines the nanometer resolution implantation of HIM with the ion-bombardment-enhanced etching (IBEE) to overcome the challenge of applying HIM in silicon nanofabrication. Since this approach owns several unique etching capabilities, we name it helium ion bombardment enhanced etching (HIBEE) for abbreviation. We experimentally verified the phenomenon reported previously [28,29,30] that the single-crystal Si lattice was damaged and converted into amorphous Si after dosing with helium ions, which exhibited a much higher etching rate than that exhibited by non-irradiated Si. This effect can be utilized to remove arbitrary shapes of helium-ion-irradiated Si without the use of a resist.

In addition, because of the unique raindrop shape of the interaction volumes of helium ions and Si, buried Si nanofluidic channels can be constructed with only one dosing step followed by one conventional chemical etching step. Moreover, suspended nanobeams of Si can be fabricated without an extra undercut step for release owing to the unique raindrop shape. Even more notable is the capability and ease of direct nanopatterning on Si micro-nanostructures with excellent resolution and accuracy of alignment due to the in-situ imaging capability of HIM. This technique can be extended to assist the etching of other difficult-to-etch materials, such as SiC, thereby creating new possibilities in nanofabrication.

## 2. Results

It has been reported that proper ion-irradiated Si can be etched away using a suitable mixture of HF and H_2_O_2_ termed IBEE [28,29,30]. This etching is attributed to the disordering of the lattice structures caused by the bombardment in which the chemical activity of the surface atoms of Si is increased.

Figure 1a shows the HIBEE process at the micrometer scale. The Si irradiated by focused helium ions changes from a crystalline lattice to an amorphous state that can be etched away by a mixture of HF and H_2_O_2_ with a much higher etching rate than that of non-irradiated Si. Petrov et al. reported that helium ion beam treatment increases the wet etching rate of silicon dioxide by a factor of five [31] and that of silicon nitride by a factor of three [32]. Herein, we demonstrated the enhanced etching of single-crystal Si. Figure 1b shows a helium ion microscope image of an array of square holes prepared using HIBEE. Arbitrary shapes can be fabricated by defining the region of helium ion irradiation. The depth was approximately 40 nm, and the surface roughness was approximately 2.1 nm for the structures produced by HIBEE, as indicated by atomic force microscopy (AFM) topography measurement. The depth of 40 nm here is due to the short etching time and is not the maximum depth which is determined by the helium ion energy. The root mean square roughness for a non-irradiated area is 1.1 nm before etching and 2.1 nm after etching while the root mean square roughness for irradiated area is 1.7 nm before etching and 2.8 nm after etching.

The increased etching rate for irradiated Si is attributed to the helium ions breaking the Si-Si bond and inducing many types of defects, including the dangling bonds of Si atoms, thereby increasing the rate of bonding of fluorine to Si and facilitating the etching process [31,33]. At the end of the ion range, each projectile creates a damage cluster that can be approximated to be a sphere of amorphous material. If the ion dose is sufficiently large, these clusters tend to overlap and produce a heavily disordered region so that Si undergoes a transition from a single crystal lattice to amorphous Si [26]. In these regions of Si, the chemical activity of the Si atoms increases so that they are more likely to be etched away by the HF-based etchant [34].

A higher etching selectivity can reduce etching time while ensuring an improved etching profile and smaller etching for the non-irradiated area. According to the study by Yonehara et al. [34], the etching selectivity between amorphous and single-crystal Si for the mixture of HF and H_2_O_2_ is thousands of times higher than that of only HF, which was similar to what we found in our experiment. Therefore, we only employed a mixture of 40% concentration of HF and 32% concentration of H_2_O_2_ in a volume mixing ratio of 1:5. We performed etching selectivity tests for different HF/H_2_O_2_ ratios and experimentally identified that at an HF/H_2_O_2_ volume ratio of 0.1 can render a selectivity as high as 210, as shown in Figure 1d. This large value of selectivity significantly exceeds previously reported selectivity values obtained using helium ion irradiation [31,32]. Petrov et al. reported a selectivity of five for silicon dioxide [31] and three for silicon nitride [32]. Additionally, at an HF/H_2_O_2_ mixing ratio of 0.1, the etching rate of the irradiated Si reached 170 nm/min. Lower or higher mixing rates resulted in lower etching selectivity and lower etching rates.

Surface changes occur at the influence values used due to sputtering or swelling, but the heights are negligible with respect to the range of the ions. The silicon under the photoresist patterns was protected from etching, while the bare silicon reacted with the fluoric and oxidative etchant and was slightly etched away so that silicon humps were formed in the same shape as the photoresist patterns. The height of the humps represented the etching depth of non-irradiated silicon, which could be measured by AFM. Moreover, the areas implanted by helium ions appeared hollow after the wet etching process and were much deeper than the non-irradiated area, indicating a faster etching rate. The etching selectivity was acquired by dividing the etching depth of the implanted area denoted as H_1_ by that of the non-irradiated area denoted as H_2_.

When the helium ion irradiation area is below the micrometer scale, the interaction volume between the helium ions and single-crystal Si begins to play a crucial role. The raindrop shape is due to the energy loss and scattering of the ions, as can be seen in Figure 2. At low ion energy, the interaction region exhibits a more circular shape; at high ion energy, the ion path governed by electronic collisions lengthens and exhibits more like a raindrop shape [19]. We used the stopping and range of ions in matter (SRIM) model [35] to simulate the single-dot dosing with the acceleration voltages and obtained simulated profiles shown in the bottom row of Figure 2a that are in good agreement with the experimental results shown in the top row of Figure 2a.

Within the interaction volumes, the Si lattice was damaged and easily etched away. Because of the raindrop shape, the etched structures had a thin neck on top and a large spherical shape at the bottom. As shown in Appendix A, the vertical distance from the cavity center to the surface is denoted as *d*_c_, the maximum depth of the cavity is denoted as *d*_max_, the lateral radius of the cavity is denoted as *R*_lateral_, and the longitudinal radius of the cavity is denoted as *R*_long_. The lateral and longitudinal radii of the cavity corresponded to the lateral and longitudinal straggling in the simulation, respectively. The maximum depth of the cavity was equal to the sum of the project range and the longitudinal straggling. For a helium ion energy of 30 keV, the project range was calculated to be 282 nm, in close agreement with the measured *d*_c_ of 294 nm. Moreover, the simulated lateral and longitudinal straggling values were 99.8 and 106.5 nm, matching well with the measured lateral and longitudinal cavity radii of 105 and 94 nm, respectively.

When the ion energy decreases, the incident helium ion is more likely to stop at a smaller vertical distance because of its collision with the Si lattice. Thus, the subsurface damage approaches closer to the surface, and the straggling dimensions shrink. Figure 2b shows the simulated geometry parameters (projected range, lateral straggling, longitudinal straggling, and total depth) of the interaction volume in Si, as well as the measured cavity geometries (*d*_m_, *d*_c_, *R*_long_, and *R*_lateral_) with increasing ion energies of 15, 20, 25 and 30 keV. This indicates that the simulation and experimental results are consistent, further suggesting that the HIBEE method is based on the selective etching of the helium ion-irradiated Si.

In single-crystal Si, increasing the number of incident ions increases the Si subsurface damage. Once the number of defects reaches a certain threshold, the crystalline structure of the substrate is completely disrupted and becomes amorphous [36]. Figure 3 shows the HIBEE experimental results exhibiting an increase in the lateral radius of the produced structures with increasing influences at the same energy of 30 keV. With increasing influences, more helium ions were laterally scattered to disrupt the lattice, resulting in a larger degree of amorphization and, thus, a higher etching rate. However, the depth of the cavities is primarily determined by the acceleration voltage, as shown in Figure 2.

The acceleration voltages and doses of helium ions determine the interaction volumes and crystal damage of single-crystal Si. The etching solvents and etching times determine the final structures. Figure 4a shows the SEM images of the nanostructures prepared with 2, 3, 5, 8, and 20 min of etching using a mixture of HF and H_2_O_2_ at a volume ratio of 1:5. Figure 5b shows that as the etching time increased, both the line width (neck width) and the lateral width increased almost linearly, but at different etching rates. The linewidth (neck width) increased at a higher etching rate of ~25 nm/min than the lateral width, for which the etching rate was ~20 nm/min. Therefore, at 20 min, the top neck had almost the same width as the bottom part. The depth of the nanostructures increased very slowly from 280 to 360 nm.

The unique raindrop-shaped cavity produced by HIBEE can be employed to fabricate buried nanofluidic channels, as illustrated in Figure 5a. Single-crystal Si was first irradiated by helium ions using a line dose comprising a single beam spot exposure gapped by 2.5 nm. Thereafter, the amorphous Si was etched away by H_2_O_2_ + HF etchants to create a tunnel with a raindrop-shaped cross-section. Due to the neck of the raindrop shape, the tunnel has a narrow opening to the surface but a larger internal diameter in the subsurface region. Finally, an extra step of atomic layer deposition seals the open neck and results in an enclosed buried channel structure that is preferred in many nanofluidic applications because it does not require an additional bonding or sacrificial layer removal step, both of which can lead to problems, such as leaking or clogging. Figure 5b shows the cross-sectional SEM images of the Si substrate after He^+^ irradiation, H_2_O_2_ + HF etching, and atomic layer deposition (ALD) sealing.

Figure 5c shows a magnified SEM image of the sealed and buried Si nanofluidic channel fabricated using HIBEE. With HIBEE, one irradiation step and one wet etching step can render a raindrop shape suitable for sealing with ALD. Conventionally, this requires the use of two etching steps, with the first etching step opening the etch window for the top neck and the second creating the pipe shape for the bottom [37]. By adjusting the parameters in HIBEE, such as the dose and energy of the helium ions and the wet etching time, the shape, diameter, and subsurface location of the nanofluidic channels can be changed, as shown in Figure 5d.

One advantage of HIBEE is the ease of defining a line at a precise position that allows the etching of a nanofluidic channel. This eliminates the alignment step required for conventional electron beam lithography. Figure 6 shows a process that combines regular microfabrication with HIBEE to fabricate nanofluidic channels with various shapes connecting two microfluidic channels.

Moreover, we demonstrated the fabrication of nanofluidic channels directly on suspended beams such as an AFM probe, as shown in Figure 7. Four microfluidic reservoirs of 30 μm × 30 μm size and 1 μm depth were produced via direct etching via Ga^+^ ions incorporated in HIM, and the nanofluidic channels were produced by HIBEE. The cross-section of the nanofluidic channel is also a raindrop shape, as shown in Figure 5c, enclosed by ALD using the aforementioned approach. This microfluidic reservoir, combined with nanofluidic channels connecting reservoirs to the end of the AFM tip, can potentially supply ink for the dip pen nanolithography (DPN) applications. The four individual reservoirs and nanofluidic channels can supply four different inks for DPN.

We can further exploit the raindrop shape of the interaction volume to achieve an undercut, which is essential for producing suspended nanobeams for NEMSs, such as nano-cantilevers, nano-switches, and nano-resonators. Figure 8a shows the fabrication process using HIBEE. Starting with a single-crystal Si wafer (step I), we performed multiple line dosing close to one another, with a spacing of typically less than 200 nm (steps II and III). The amorphous Si on the bottom under each line partially overlapped underneath the surface, while the amorphous Si on the top neck retained its individuality. Thereafter, we used an etchant of HF and H_2_O_2_ to etch away the amorphous region and immediately render suspended Si nanobeams. Figure 8b–d show the SEM images of an array of linear nanobeams, a suspended micro mesh comprising cross-bar nanobeams, and a suspended serpentine nanobeam. Thus, we demonstrated the fabrication of suspended nanobeams in one single-crystal Si etching step without the use of sacrificial layers.

During the release of suspended nanobeams, surface tension tends to damage the beam if the beam is long and thin. In our demonstrated nanobeams, we chose the length of the beam no more than 10 times the beam width. If longer beams are needed, then a super critical drying process would be required to assist the successful release of suspended beams, which is a standard process used for releasing high aspect ratio beams in the MEMS/NEMS field [38].

## 3. Discussion

HIBEE is a simple, fast, effective method for the fabrication of single-crystal Si nanostructures. It solves a critical issue of helium ion milling that results only in swelling and renders it almost impossible to produce high-quality Si nanostructures by the milling of single-crystal Si [27]. Thus, it provides a feasible solution for the fabrication of Si nanostructures using focused helium ion microscopes. Moreover, HIBEE has several distinctive advantages for the production of NEMS devices.

First, in HIBEE, helium ions are directly implanted into the Si substrate to define the area to be removed. A single etching step was sufficient to remove the implanted areas. Therefore, HIBEE can be easily integrated with existing manufacturing processes, such as adding nanostructures to an existing device, as shown in Figure 6. Moreover, unlike conventional electron beam lithography or ion beam lithography, HIBEE does not require a special resist and resist spin coating step, aiding the production of nanostructures on many surfaces that were previously unsuitable for nanostructure fabrication. For example, HIBEE can produce nanostructures on complicated surfaces, including 3D surfaces, for which spin coating is impossible, and EBL is difficult to perform. Figure 7 shows that HIBEE can directly produce multi-nanofluidic channels on a 3D Si pyramidal tip, as well as on the suspended Si cantilevers of an atomic force microscopic probe, which has considerable potential to provide inks via nanofluidic channels for continuous dip pen lithography [39]. As shown in Figure 7, four self-enclosed nanofluidic channels can provide four different molecular inks for supplying the tip without exposure to air that causes ink degradation. The use of such nanofluidic channels also eliminates cross-contamination even when the channels are very near one another. Other potential applications include the fabrication of nano-fountain pens for molecular writing [40], the fabrication of nanofluidic channels on top of a suspended nano-resonator for attogram mass measurements [41], and single-cell growth monitoring [42].

Second, it allows suspended nanostructures to be easily produced without sacrificial layers. For example, in conventional micro/nanofabrication, to produce a suspended beam, a sacrificial layer is typically required to release the suspended structures, or, alternatively, SOI wafers can be used [43]. For SiC-suspended nano-resonators, Si is typically etched to release the suspended SiC nanobeams [44]. However, due to the intrinsic raindrop interaction volume of helium ions and single-crystal Si, we can use two line doses sufficiently near one another to produce a suspended nanobeam with a quasi-triangular cross-section.

Third, it provides a feasible route for etching materials that are typically difficult to directly etch away by wet etching or even dry etching. For example, SiC is almost chemically inert and, therefore, difficult to wet etch. Only plasma-based reactive ion etching can be used for SiC etching [45]. However, inductively coupled plasma reactive etching (ICP-RIE) and micromasking can significantly damage the sidewalls and surfaces [46,47]. In addition, RIE can cause an increase in the number of trap states in SiC [48], preventing its use in applications that require a low interface defect density. Herein, we also demonstrated the HIBEE of SiC, as shown in Appendix A. A series of rectangular micro-holes were fabricated by HIBEE, incorporating the etching process using a mixed etchant of HFand H_2_O_2_ heating at 80 °C. Appendix A shows that the roughness of the bottom surface was 0.753 nm, while the depth of the square hole was 217 nm.

One limitation of HIBEE is that the penetration depth is approximately hundreds of nanometers, which is due to the restrictions imposed by the helium ion energy that is limited to approximately 30 keV of the helium ion microscope. To achieve deeper structures, multiple steps of dosing and etching can be implemented, similar to a reported method using multiple steps of gallium ion implantation and etching for 3D nanostructure fabrication [49]. Alternatively, nanoscale masks can be fabricated on the substrate, followed by the use of higher-energy helium ions with energies up to several MeVs produced using a commercial ion implanter [50].

## 4. Conclusions

We reported a straightforward nanofabrication approach that exhibits several unique nanofabrication capabilities. We optimized the mixing ratio of the HF and H_2_O_2_ etchants and obtained a high selectivity of 210. Thereafter, we demonstrated buried nanofluidic channel fabrication using merely one dosing step and one etching step in the range of 50 to 250 nm of linear channels, as well as serpentine and arbitrary shapes. Moreover, we showed that it could be readily employed to fabricate arbitrary shapes of nanostructures on 3D surfaces without the use of a photoresist that is required in conventional EBL. This approach opens up many exciting avenues in nanotechnology applications, such as mechanical nano-resonators, nanofluidic channels, and dip pen nanolithography.

## 5. Experimental Section

P-type <100> single-crystal Si wafers with a resistivity in the range of 1–30 Ω·cm^−1^ were used in this study. Helium ion implantation was performed using HIM (ZEISS, Orion Nanofab) with feature sizes in the range of 5–10 nm. The typical ion energy was 30 keV, whereas, in Figure 3, it was varied from 15–30 keV to investigate the effect of ion energy on the penetration volume. Thereafter, the samples were etched in the solutions containing various amounts of H_2_O_2_ (40%) and HF (32%) for 0.5–30 min to selectively remove the amorphous Si after ion implantation.

Etching selectivity tests were performed, as shown in Appendix A. First, photolithography was performed on top of a single-crystal Si wafer to produce micro-disk photoresist patterns. Then HIM was applied to implant helium ions with a dosage of 1000 ions/nm^2^ within special patterns (e.g., 2 µm × 2 µm rectangles) on bare silicon regions not covered by photoresist patterns. Afterward, the wafer was etched by mixtures of HF and H_2_O_2_ and then rinsed successively in deionized water, acetone, and isopropyl alcohol to remove the photoresist. In the etching selectivity test, the volume ratio of HF to H_2_O_2_ is from 0.02 to 10. In all the other fabrication experiments, an optimal ratio of 0.2 was employed.

To fabricate nanofluidic channel structures, we first used helium ions to dope the line structures. Thereafter, we used wet chemical etching to produce an undercut tube-like structure. Next, a conformal layer, such as SiO_2_, was deposited by ALD (Picosun, Sunale R-200 Advanced) to seal the seam over the tube, resulting in buried nanochannel structures within the Si substrate. For SiO_2_ deposition, SiCl_4_ and O_3_ were used as the ALD precursor molecules at a temperature of 400 °C, with a deposition rate of 0.4 Å/cycle.

For the nanofluidic experiment, a microfluidic device with two isolated channels was first fabricated using a traditional semiconductor manufacturing process, including photolithography and dry etching. The critical size of the micro-channels was 100 μm × 100 μm (width × depth). The microchannels were arranged parallel to one another at a distance of 40 μm. Subsequently, nanofluidic channel structures were fabricated using HIBEE to connect the two microfluidics. To enhance the hydrophilicity of the channels, 10 min of oxygen plasma etching was performed to render the Si surface hydrophilic and facilitate the entry of the fluorescence solution into the nanofluidic channel.

## Figures and Tables

**Figure 1 nanomaterials-12-03269-f001:**
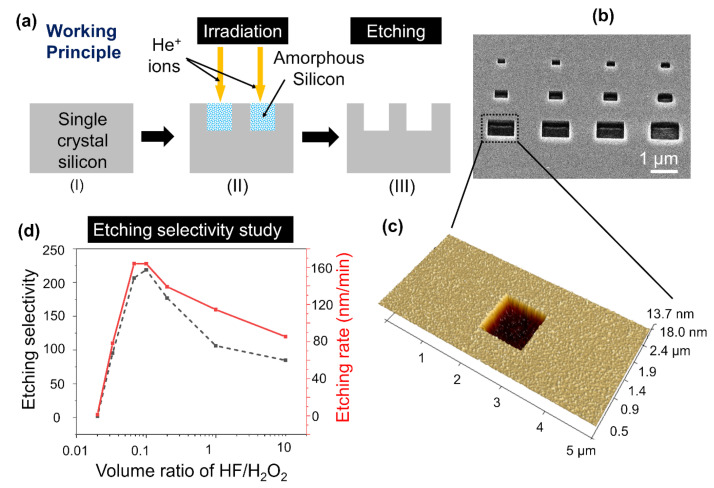
(**a**) Principle of helium ion-bombardment enhanced etching for fabricating structures at the micrometer scale: (I) start with single-crystal Si, (II) irradiate with focused helium ions, (III) etch away amorphous Si formed by helium ion irradiation. (**b**) an SEM image of an array of Si square holes of different sizes fabricated by HIBEE. (**c**) AFM topography measurement of an etched square hole. (**d**) Etching selectivity of irradiated Si over non-irradiated Si, as well as the etching rates of irradiated Si for different HF/H_2_O_2_ mixing ratios.

**Figure 2 nanomaterials-12-03269-f002:**
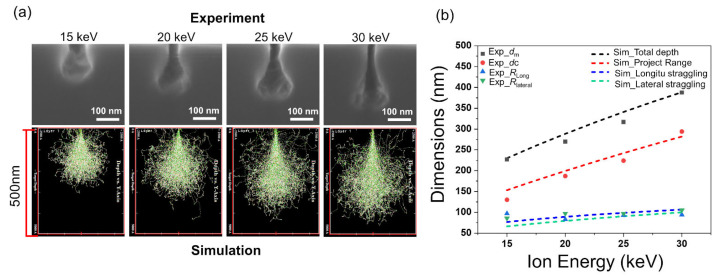
(**a**) Experimental(upper row) and simulation(lower row) results of Si nanostructures fabricated by HIBEE with different acceleration voltages of helium ions of 15, 20, 25, and 30 keV. (**b**) The experimental and simulation dimensions of the ion dispersion range with different ion energies, both indicating that higher ion energy leads to deeper channels and larger lateral straggling.

**Figure 3 nanomaterials-12-03269-f003:**

SEM images of the etched Si nanostructures using HIBEE with the same energy by increased influences of helium ions. A larger influence leads to a larger percentage of amorphization and thus more etching; however, the maximum depth of the cavity is almost constant.

**Figure 4 nanomaterials-12-03269-f004:**
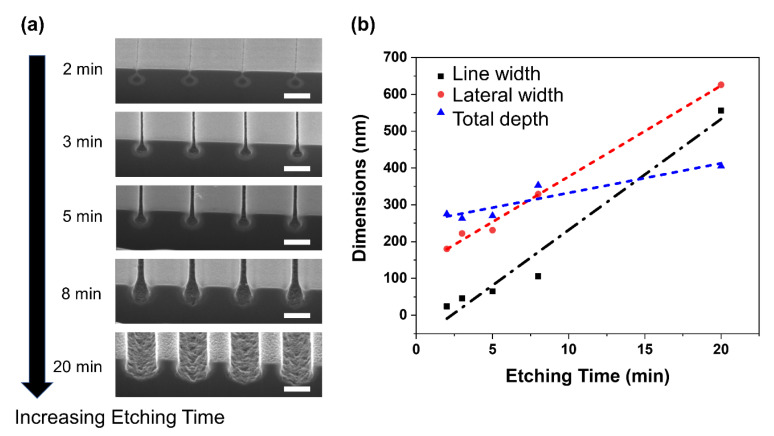
Si nanostructures produced by HIBEE using 2, 3, 5, 8, and 20 min of wet etching time. (**a**) SEM images. (**b**) Plots of line width, lateral width, and total depth of the nanostructures. The scale bar is 500 nm.

**Figure 5 nanomaterials-12-03269-f005:**
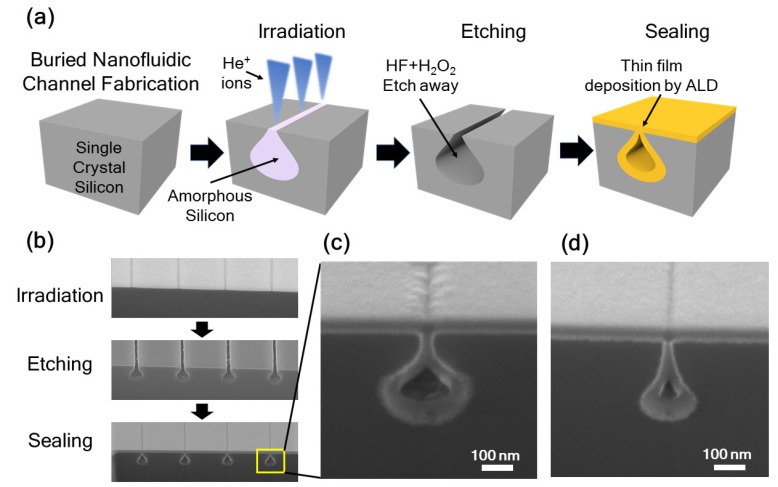
(**a**) Schematics of the process of fabricating buried Si nanofluidic channels using HIBEE. (**b**) SEM images of single-crystal Si undergoing helium ion beam line irradiation, etching, and atomic layer deposition of SiO_2._ (**c**) Zoomed-in cross-sectional view of a sealed and buried Si nanofluidic channel produced by a rectangle dose of 10 µm × 10 nm with an influence of 31,205 ions/nm^2^. (**d**) Zoomed-in cross-sectional view of a sealed and buried Si nanofluidic channel with a smaller diameter and a different shape obtained by tuning the HIBEE processing parameters. A rectangle dose of 10 µm × 10 nm area size with an influence of 6241 ions/nm^2^ was employed.

**Figure 6 nanomaterials-12-03269-f006:**
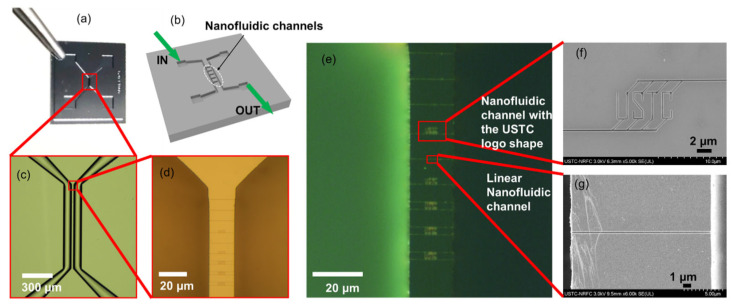
(**a**) Photograph of the fabricated micro-nanofluidic channel device. (**b**) 3D cartoon image illustrating the micro-nanofluidic device. (**c**,**d**) Bright-field optical microscopic photographs of the micro-nanofluidic channels. (**e**) Fluorescence microscopic image showing the fluorescence flowing from the left-side microchannel to the right-side microchannel through the nanofluidic channels, confirming the functionality. (**f**) SEM image of a customized nanofluidic channel with the USTC logo shape. (**g**) SEM image of a linear nanofluidic channel.

**Figure 7 nanomaterials-12-03269-f007:**
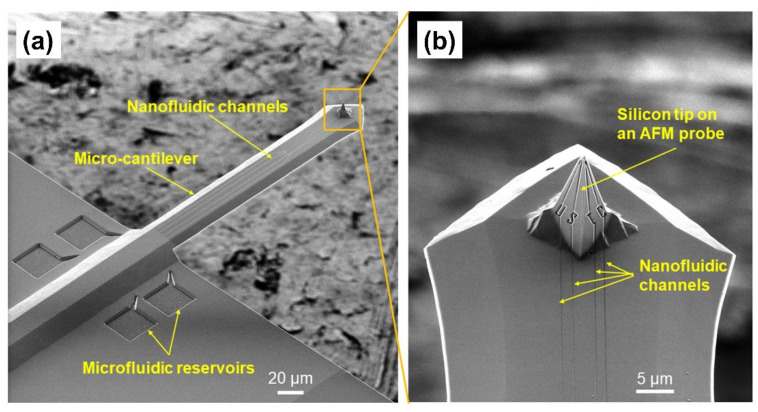
(**a**) Nanofluidic channels directly produced on a suspended micro-cantilever and a Si tip on top of an atomic force microscopic probe. (**b**) Zoomed-in view of the Si nanofluidic channels on both the micro-cantilever and the Si tip.

**Figure 8 nanomaterials-12-03269-f008:**
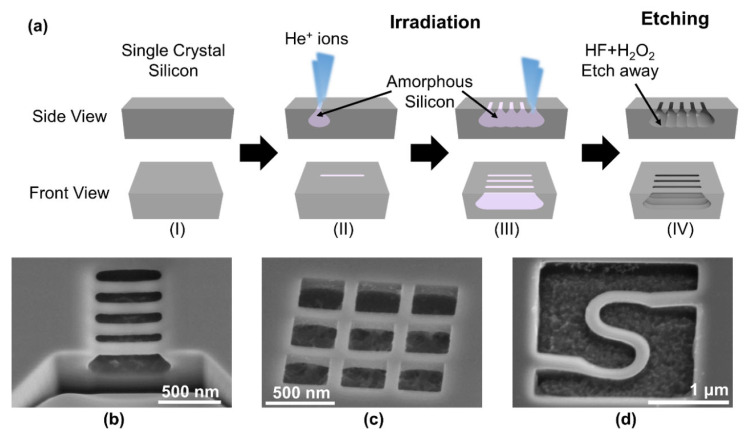
(**a**) Fabrication process of suspended nanobeams using HIBEE including starting from a single crystal silicon substrate (step I), helium ion irradiation (step II & III), and wet etching (step IV). SEM images of (**b**) an array of suspended straight nanobeams, (**c**) a suspended Si nano-mesh, and (**d**) a suspended serpentine Si nanobeam.

## Data Availability

Not applicable.

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
