# Peer review of "Versatile Approach of Silicon Nanofabrication without Resists: Helium Ion-Bombardment Enhanced Etching"

_nanomaterials, 2022, doi:10.3390/nano12193269_

Round 1

Reviewer 1 Report

The article submitted for review titled “Versatile Approach of Silicon Nanofabrication without Resists: Helium Ion-Assisted Chemical Etching (HiaEtch)” presents a method for chemical etching of silicon substrates without the use of any type of lithography method by employing a high energy helium ion beam. In essence, the high energy ion beam is used to “pattern” the crystalline silicon substrate, disrupting the crystalline lattice in the irradiated areas and increasing the etch rate of these areas by a factor of 200, compared to the non-irradiated areas.

The authors use the SRIM model to simulate the interaction between He ions and the substrate in function of acceleration voltages and compare the simulation results with experimental results showing the dependency between ion energy versus etch depth and cavity shape.

Furthermore, the authors present several possible applications for the proposed nanofabrication method, by employing the unique tear-drop shape generated by the HiaEtch method: fabrication of fully enclosed nanofluidic channels sealed by a subsequent ALD deposition; fabrication of suspended Si beams.

The paper presents a novel nanofabrication method, and is well written. I suggest the editors to publish the paper as is.

Reviewer 2 Report

see pdf

Reviewer 3 Report

The reported content and results are understandable and the paper is interesting from an engineering perspective.However, as a scientific paper, the results and discussion are unclear.

Results section 2 should clearly state the various experimental conditions. It is acceptable to be in the figure caption of each figure. 

In the Discussion section 3, the results shown in Fig. 8 do not appear in the results section 2 at all, they appear for the first time and are not appropriate for a scientific paper structure. Please describe them in the results section 2. The paper would be more interesting if you provide details in the results section 2 and discuss, for example, the size, morphology of the nanofluidic channels in the figure 8.

In the experimental section 5, you need to describe how you controlled the beam spot size of 5 - 10 nm (line 317) in the helium ion beam irradiation conditions. 0.5 nm diameter is published as Orion Nano-Fab's beam diameter. Does this mean that the beam diameter had to be increased for this experiment? Please explain.

For Conclusion Session 4, please be a little more specific, and specify what new size and shape you have been able to achieve with this technology.

Reviewer 4 Report

The authors reports on a rather interesting technique that is sure to resonate in the scientific community. Overall, the paper is well written and conveys the core scientific aspects.

Some points are open:

- please increase the font size in Fig. 1d, also true for other figs.

- in line 91 it sounds that HF and H2O2 were also used individually and not as a mixture - can you please comment on this

- please add more information in the text about the H2O2 and HF concentration types (e.g. BHF vs. HF) and their concentrations, otherwise the ratios are not clear

- suspended/freestanding nanostructures like your nanoantilevers tend to be damaged by surface tension forces when pulled out of an aqueous solution. Have you encountered such problems and how did you solve them?

Round 2

Reviewer 2 Report

see attachement
